# Brief COPE Short Version (Mini-COPE): A Proposal of Item and Factorial Reduction in Mexican Population

**DOI:** 10.3390/healthcare11081070

**Published:** 2023-04-08

**Authors:** Aniel Jessica Leticia Brambila-Tapia, Reyna Jazmin Martínez-Arriaga, Joel Omar González-Cantero, Victor Hugo González-Becerra, Yesica Arlae Reyes-Domínguez, María Luisa Ramírez-García, Fabiola Macías-Espinoza

**Affiliations:** 1Departamento de Psicología Básica, Centro Universitario de Ciencias de la Salud (CUCS), Universidad de Guadalajara, Guadalajara 44340, Mexico; 2Departamento de Clínicas de Salud Mental, Centro Universitario de Ciencias de la Salud (CUCS), Universidad de Guadalajara, Guadalajara 44340, Mexico; 3Departamento de Ciencias del Comportamiento, Centro Universitario de los Valles (CUVALLES), Universidad de Guadalajara, Guadalajara 46600, Mexico; 4Maestría en Psicología de la Salud, Departamento de Psicología Básica, Centro Universitario de Ciencias de la Salud (CUCS), Universidad de Guadalajara, Guadalajara 44340, Mexico; 5Departamento de Psicología Aplicada, Centro Universitario de Ciencias de la Salud (CUCS), Universidad de Guadalajara, Guadalajara 44340, Mexico

**Keywords:** brief COPE, factorial analysis, reduced version, Spanish-speaking populations, Mini-COPE

## Abstract

The factorial reduction of Brief COPE has not been successfully replicated by independent studies, and few have been performed in Spanish-speaking populations; therefore, the objective of this study was to perform a factorial reduction of the instrument in a large sample of the Mexican population and perform a convergent and divergent validity of the factors obtained. We distributed a questionnaire via social networks with sociodemographic and psychological variables, including the Brief COPE and the scales of the CPSS, GAD-7, and CES-D to measure stress, anxiety, and depression. A total of 1283 persons were included, most of whom (64.8%) were women and had a bachelor’s degree (55.2%). After performing the exploratory factorial analysis, we did not find a model with an adequate fit and a reduced number of factors; therefore, we decided to reduce the number of items according to the most representative ones of adaptive, maladaptive, and emotional coping strategies. The resulting model with three factors showed good fit parameters and good internal consistency of the factors. In addition, the nature and naming of the factors were confirmed by convergent and divergent validity, with significant negative correlations between factor 1 (active/adaptive) and stress, depression, and anxiety, significant positive correlations between factor 2 (avoidant/maladaptive) and these three variables, and no significant correlation between factor 3 (emotional/neutral) and stress or depression. This shortened version of the brief COPE (Mini-COPE) is a good option to evaluate adaptive and maladaptive coping strategies in Spanish-speaking populations.

## 1. Introduction

Coping strategies are defined as an individual’s attempts to use cognitive and behavioral strategies to manage and regulate pressures, demands, and emotions in response to stress [1]. Stress is considered a “particular relationship between the person and the environment that is appraised by the person as taxing or exceeding his or her resources and endangering his or her well-being” [2]. So far, coping strategies have been classified into different types according to authors and instruments employed; for instance, the most commonly used instruments are the coping strategies inventory (CSI) [3,4] and the Brief COPE [5,6]. Although these two instruments are similar and share the assessment (measurement) of common strategies, the Brief COPE is very commonly used in clinical and health contexts [7,8].

The Brief COPE measures a higher number of coping strategies (14 vs. 8). The Brief COPE consists of 28 items that measure 14 different subscales (including: self-blame, behavioral disengagement, self-distraction, denial, substance use, emotional support, instrumental support, active coping, planning, acceptance, positive reframing, religion, venting, and humor), with two items for each subscale. The Brief COPE has been reduced to different dimensions depending on the study. For example, Cooper et al. reduced the instrument into three dimensions: emotion-focused strategies, problem-focused strategies, and dysfunctional-focused strategies [9]; and Meyer et al. classified it into two big dimensions: adaptive and maladaptive coping strategies [10]. However, none of these reports has been validated with the factorial analysis. To date, many studies have tried to reduce the number of factors in the Brief COPE using exploratory factor analysis (EFA) and/or confirmatory factor analysis (CFA) in different clinical and general populations [8]. However, most of them have been performed with small sample sizes (less than 500 subjects) and have yielded a similar number of factors as the original instrument, making the factorial reduction useless in practice. In addition, most of the factorial reductions among the different populations studied are not similar. For instance, Su et al. [11] reduced the instrument with EFA and CFA to six factors in an HIV population, while Baumstarck et al. [12] reduced the instrument to four different factors in cancer patients and their caregivers in the French population, with a few coincidences in the item arrangement for each factorial structure in both studies. In addition, only four reports of factorial reduction of the Brief COPE have been proposed in Spanish-speaking populations: one in Argentina, two in Spain, and one in Mexico, where the authors used EFA or CFA and found great variability in the number of factors: from 2 to 14, showing discordances even within the same population. By way of example, the two different studies performed in Spain yielded 14 and 12 factors in the EFA [13,14], and the only study performed in a Mexican population of breast cancer patients [15] yielded 7 factors of the original 14, leaving intact 5 original subscales: humor, self-blame, substance use, self-distraction, and planning.

Among the variables that can modify the factorial structures observed in the different populations are: (a) The different frequencies and combinations of coping strategies used in each studied population, and (b) The type of coping being measure with the Brief COPE (situational or dispositional), this by considering that the instrument varies in its redaction, and in some studies it measures situational coping strategies (state and trait oriented, i.e., “I criticize myself”) while in other studies it measures dispositional coping strategies (which identifies coping skills utilized during an specific period of time, i.e., “I have been criticizing myself”), which, although they are expected to be related, they represent different measurements [16]. In this study we utilized the situational Brief COPE.

Considering these discrepancies in the factorial reduction of the instrument, it is fundamental to perform more studies that intend to reduce the Brief COPE into fewer factors, which can be easily applicable in different health and general contexts, even if this reduction implies some item reduction. In this sense, it is important to mention that although the Brief COPE is already reduced from the original full COPE instrument of 60 items [17], its analysis is complex when considering that it has 14 subscales, many of them belonging to similar coping styles, like avoidant or active coping. Therefore, its reduction into fewer factors, which represent the type of coping they are measuring, could be very helpful in order to make the instrument more useful and practical. The objectives of this study are: (1) to reduce the brief COPE into functional dimensions, even if this implies an item reduction, in a sample of the Mexican general population; and (2) to corroborate the factorial structure obtained with convergent and divergent validity in order to identify the type of coping that each factor represents. For this last objective, we measured three psychological scales of stress, anxiety, and depression, expecting that a factor that represents maladaptive coping strategies would show significant positive correlations with them while a factor representing adaptive strategies would show a negative correlation with them.

## 2. Subjects and Methods

The target population was the Mexican adult population, which was reached with the snowball sampling method, in which the research team distributed an electronic questionnaire with sociodemographic and psychological instruments by social networks including WhatsApp, Facebook, and e-mail. In this sense, most participants were university students, which represents mainly young and educated people. The instrument was designed by the research team, and its understanding was tested on a small number of students (the first participants).

The study was approved by the ethics and research committee of the Health Sciences University Center of the University of Guadalajara (registration number: CI-06821), and the participants gave their consent to participate in the same questionnaire.

The socio-demographic data included sex, age, whether they have a romantic partner, schooling, whether they have a job, and socio-economic level, which was measured in five possible categories: from very low to very high.

The psychological measures included the coping strategies, measured with the brief-COPE scale [5,6] with a range of 0–3, and for this instrument, we used each item to perform the factorial reduction. In addition, we obtained the average of the punctuation obtained in the items included in each factor in order to perform the correlations with the other psychological scales (stress, depression, and anxiety). To identify the convergent and divergent validity of the instrument, we measured: stress with the Cohen Perceived Stress Scale (CPSS) [18,19], with a range of the instrument of 1–5; depression, measured with the CES-D Scale [20,21], with a range of the instrument of 0–3; and anxiety, measured with the GAD-7 Scale [22,23] with a range of the instrument of 0–3. These three instruments have also been validated in the Spanish-speaking population, and they only have one whole scale per instrument, which means that they are not divided into subscales. For analysis purposes, we also obtained the average of the punctuation obtained in the items conforming to each scale for each participant.

### Statistical Analysis

To describe qualitative variables, we used frequencies and percentages, and for quantitative ones, means and standard deviation. To determine the EFA and CFA, we used the JASP software [24]. In the EFA, we obtained the factor load using the orthogonal varimax rotation analysis, assuming that factors are correlated, and the estimator method was the minimum residual. For the CFA, we obtained the comparative fit index (CFI) with the maximum likelihood (ML) estimator method; we also obtained the Tucker-Lewis index (TLI), the standardized root mean square residual (SMRS), and the root mean square error of approximation (RMSEA) values. Values of CFI and TLI ≥ 0.90 and SMRS and RMSEA ≤ 0.08 are considered good fits for the model [25]. In order to determine the convergent validity of the factors obtained with the confirmatory factorial analysis, we used the Spearman correlation test (considering the non-parametric distribution of the data) to compare each factor with stress, depression, and anxiety. We also obtained the Cronbach’s alpha tests for all the instruments employed and factors obtained from the Brief COPE; a value ≥ 0.60 was considered acceptable [26]. These statistical analyses were performed with the SPSS v.25 software.

## 3. Results

We verified that all questionnaires were congruently filled out in order to discard false information. A total of 1283 participants older than 18 years old were included, of whom 64.8% were women. The mean ± SD of age was 31.42 ± 11.28 years. Most of the studied population had a romantic partner (61.6%), had a job (67.3%), had a bachelor’s degree (55.2%), and had a medium socioeconomic level (81.2%); this refers to the social and economic position that a person has. The Cronbach’s alpha test for the CPSS scale was 0.855; for GAD-7, 0.923; and for CES-D, 0.867.

### 3.1. EFA and CFA

The EFA analysis gave a result of six potential factors; however, three of these factors were made up of two items each (corresponding to the subscales of substance use, humor, and religion), and six items were not clearly integrated into any factor. Finally, when we performed the confirmatory factorial analysis with this item distribution, the CFI value was barely acceptable (<0.91). We additionally performed the confirmatory factorial analysis for the factor structure proposed by Cooper and Meyer [9,10] and by the reports in China and France [11,12], but the fitness of the model in neither case was acceptable (with CFIs of 0.490, 0.366, 0.795, and 0.613, respectively).

### 3.2. Item Reduction and CFA

We then performed an item reduction, leaving only the most representative subscales for active/adaptive strategies (4 items: active coping and planning), avoidant/maladaptive strategies (4 items: denial and behavioral disengagement), and emotional/neutral coping strategies (4 items: emotional and instrumental support), with 12 items in total. This selection was performed based on the results observed in the EFA, together with the theoretical analysis of these items as being the most representative of a specific coping style. The minimum item loading was 0.45, with no item overlap in two factors (Table 1). With this factor structure, we obtained a good fit for the model (CFI = 0.954) (Table 2). Additionally, the Cronbach’s alpha values of the three factors were high for factors 1 and 3 (>0.80) and acceptable for factor 2 (>0.60) (Table 2). Considering the important item reduction, we propose a different name for this version: brief COPE short version (Mini-COPE). Very similar fit parameters were obtained if only factors 1 and 2 were included in the model, with a CFI of 0.954, TLI of 0.933, RMSEA of 0.067, and SRMS of 0.040 (Table 2).

Finally, we performed correlations between each of the three factors obtained and the values of stress, depression, and anxiety in the studied population. In these analyses, we confirmed the convergent and divergent validity of each factor (Table 3), finding significant negative correlations between factor 1 (active/adaptive coping strategies) and the three negative psychological variables; we also found significant positive correlations between factor 2 (avoidant/maladaptive coping strategies) and the three psychological variables; no significant correlations between factor 3 (emotional/neutral coping strategies) and stress or depression; and only a very low positive correlation was found between anxiety and factor 3 (Table 3). These correlations, along with theory, led us to name these coping types as adaptive, maladaptive, and neutral.

## 4. Discussion

Brief COPE is a widely used instrument in the measurement of coping strategies; however, the large number of strategies that it measures and the wide variability of factorial reduction models [8,9,10,11,12,13] do not permit an easy and confident use of the instrument. This indicated the need for more studies with larger sample sizes and new proposals that can give rise to a useful and confident factorial model that is applicable to the Spanish-speaking population and that can even be replicated in different populations.

We observed after a careful factorial analysis, where EFA yielded a high number of factors similar to those observed in previous attempts to reduce this instrument, as reported in the review of Rodrigues et al. [8]; that it was possible to perform a considerable item reduction, giving three useful factors, and whose model showed an adequate fit. In addition, we could determine, by convergent and divergent validity, the names of each factor that better represent the items included. With this proposal, we reduced the number of items from 28, measuring 14 different coping styles, to 12, measuring three different coping styles: active/adaptive, avoidant/maladaptive, and emotional/neutral.

According to our second objective, we could name these factors based on the convergent and divergent validity obtained with the correlations between each factor and the values of stress, depression, and anxiety obtained in the studied population. In this sense, it is noteworthy that the factor of emotional/neutral strategies showed no significant correlations with stress or depression, and only a very low positive correlation was detected with anxiety; therefore, we decided to name this factor “neutral”. Nevertheless, it is important to consider that the very low correlations observed with stress, depression, and anxiety, reaching significance only with anxiety, could indicate that these emotional strategies (emotional and instrumental support) are slightly maladaptive; however, this should be determined with more and larger studies. The other two factors were clearly adaptive (Factor 1) and maladaptive (Factor 2) according to theory and the correlations performed. In this sense, and considering that these are the more relevant factors for clinical and research usage, it is possible to use only these factors, reducing the instrument to eight items. The model fit values did not change when only factors 1 and 2 were included.

The naming of the three factors obtained with this item reduction is similar to the dimensions proposed by Cooper et al. [9], which were emotion-focused strategies, problem-focused strategies, and dysfunctional-focused strategies. So, the originally 14 subscales proposed by Carver [5] are reduced to three functional and practical factors or dimensions that correspond to a previously proposed one and are very similar to the two big dimensions proposed by Meyer (adaptive and maladaptive coping styles) [10].

Although with this proposal many coping strategies, most of them maladaptive, were not included, the main adaptive and maladaptive strategies are included in the model. We consider that this new reduced version of the instrument can be more useful in terms of instrument application (saving more time) and mainly for the data analysis, performing comparisons only with 3 factors (equivalent to subscales) instead of the 14 different subscales of the original instrument. According to the CFA performed, the instrument can be even more reduced by including only the main two factors (Factors 1 and 2), which represent adaptive and maladaptive coping strategies, and making even faster application, considering that these factors are measured with only eight items.

In addition, by considering the good fit of the model obtained and the congruence between the items within each factor (each one representing a different type of coping strategy), we think that this new version can be easily replicated in different populations.

The advantages of this study are the large sample size, considering that larger sample sizes approximate better to the evaluated population and give more reliable results; the evaluation of previous reported models by the CFI; and the measurement of stress, depression, and anxiety to determine the convergent and divergent validity. The limitations of the study are the use of a non-random sampling method, which diminishes the representativeness of the Mexican general population; the fact that we measured mainly young and educated people, which underrepresents the older people and those from lower socioeconomic levels; and, finally, the lack of follow-up, which would have permitted us to determine the intraclass correlation coefficients.

In conclusion, we performed a new proposal of item and factor reduction of the dispositional Brief COPE (Mini-COPE) instrument in the Mexican population, where we included 12 items corresponding to 3 factors, named active/adaptive, avoidant/maladaptive, and emotional/neutral, and which showed statistical parameters indicating a good fit of the model, an acceptable internal consistency of each factor, and expected correlations with stress, depression, and anxiety in the convergent and divergent validity. This reduction is expected to help the professional community by focusing on the clinically relevant strategies (adaptive and maladaptive ones), which can be measured with a higher degree of accuracy (considering that four items are included in each factor), and by saving time on its application.

In addition, the inclusion of convergent and divergent validities sheds more light on the comprehension of the relationship between coping strategies and mental health. The performance of further studies with this proposal, including the situational version of the Brief COPE, would determine its usefulness and confirm the fitness of the model.

## Figures and Tables

**Table 1 healthcare-11-01070-t001:** Items loaded in the proposed factors (English/Spanish).

Items	Factor 1	Factor 2	Factor 3
1. I’ve been concentrating my efforts on doing something about the situation I’m in./*“Concentro mis esfuerzos en hacer algo sobre la situación en la que estoy.”*	0.654		
2. I’ve been taking action to try to make the situation better./*“Tomo medidas para intentar que la situación mejore.”*	0.714		
3. I’ve been trying to come up with a strategy about what to do./*“Intento proponer una estrategia sobre qué hacer.”*	0.771		
4. I’ve been thinking hard about what steps to take./*“Pienso detenidamente sobre los pasos a seguir.”*	0.631		
5. I’ve been saying to myself “this isn’t real”./*“Me digo a mí mismo: esto no es real”*		0.454	
6. I’ve been giving up trying to deal with it./*“Renuncio a intentar ocuparme de ello.”*		0.592	
7. I’ve been refusing to believe that it has happened./*“Me niego a creer que haya sucedido.”*		0.652	
8. I’ve been giving up the attempt to cope./*“Renuncio al intento de hacer frente al problema.”*		0.542	
9. I’ve been getting emotional support from others./*“Consigo apoyo emocional de otros.”*			0.770
10. I’ve been getting help and advice from other people./*“Consigo que otras personas me ayuden o aconsejen.”*			0.791
11. I’ve been getting comfort and understanding from someone./*“Consigo el consuelo y la comprensión de alguien.”*			0.762
12. I’ve been trying to get advice or help from other people about what to do./*“Intento conseguir que alguien me ayude o aconseje sobre qué hacer.”*			0.598

**Table 2 healthcare-11-01070-t002:** A confirmatory factorial analysis of the factorial model and internal consistency.

	Xi^2^	df	*p* Value	CFI	TLI	SRMR	RMSEA
**3 factor model**	266.27	51	<0.001	0.954	0.941	0.041	0.057
**2 factor model**	128.175	19	<0.001	0.954	0.933	0.067	0.040
**Cronbach’s alphas**							
**Factor 1 (Active/Adaptive)**	0.805						
**Factor 2 (Avoidant/Maladaptive)**	0.645						
**Factor 3 (Emotional/Neutral)**	0.837						

Convergent and divergent validity.

**Table 3 healthcare-11-01070-t003:** Bivariate correlations between each factor and stress, depression, and anxiety (convergent and divergent validity).

Factors	Stress	Depression	Anxiety
**Factor 1 (Active/Adaptive)**	−0.323 **	−0.258 **	−0.127 **
**Factor 2 (Avoidant/Maladaptive)**	0.401 **	0.444 **	0.348 **
**Factor 3** **(Emotional/Neutral)**	0.037	0.043	0.090 **

** *p* value < 0.01. *p* values obtained with Spearman correlation test.

## Data Availability

Data are available upon reasonable request.

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
