# Peer review of "Brief COPE Short Version (Mini-COPE): A Proposal of Item and Factorial Reduction in Mexican Population"

_healthcare, 2023, doi:10.3390/healthcare11081070_

Round 1
Reviewer 1 Report
Brief COPE short version (Mini-COPE): a proposal of item and factorial reduction in Mexican population aimed to reduce the Brief COPE, and by doing so simplify analysis of the instrument in Spanish-speaking populations.
There are many errors in word choice, grammar, flow, etc. The manuscript needs a careful read before moving forward. I will address some of the issues, but the list below is not intended to be exhaustive.
Abstract:
1. Lines 18-20 – It’s not clear what the objective of the study is from this sentence. Is it to reduce the factors or replicate the reduction of factors?
2. Line 33 – do you mean Spanish-speaking populations or a Spanish-speaking population?
Introduction:
1. The first sentence needs work. Line 39, which in turn, is considered… What is considered?
2. The second sentence (line 41), doesn’t make sense.
3. Line 44: Although these two instruments are similar and share assessment (or measurement) strategies, the Brief COPE is the most used in clinical and health contexts and measures a more significant number of strategies…
4. Line 46: not in
5. Line 47: depending on the study. For example, Cooper … instrument to three dimensions.
The suggestions above should give the authors an idea of the types of word choice/grammar issues that need to be addressed.
a. One major issue with the summary of existing attempts to reduce factors in the COPE is the omission of the version of the COPE (situational vs dispositional) used in the studies and the impact that might have on results. See: Solberg MA, Gridley MK, Peters RM. The Factor Structure of the Brief Cope: A Systematic Review. Western Journal of Nursing Research. 2022;44(6):612-627. doi:10.1177/01939459211012044
b. Doesn’t the inability of the studies cited to “agree” on factors indicate that there may be differences in the populations being studied and the types of strategies they might use?
Subjects and methods:
1. How were participants selected? How was socioeconomic level measured?
Statistical Analysis:
1. Read and edit for clarity and site SPSS.
Results:
1. What does a “medium socioeconomic level” mean?
2. Be consistent with using EFA and CFA.
3. Line 128: 9, 10 are the incorrect references here
4. Revise sentence lines 157-158 for clarity.
Discussion:
1. I suggest reorganizing the Discussion. Specifically, perhaps switch paragraphs 1 and 2. Also, in doing so, revise the first sentence in the paragraph beginning Brief COPE… for clarity. It is too long and difficult to follow. There are other sentences that a very long and need work.
2. The first sentence of paragraph beginning line 202 doesn’t make sense.
3. Revise the concluding paragraph. Be clear about the version of the COPE used and the conclusions that can be drawn from the results.
Other issues:
In the Introduction there is talk of replicating previous findings, yet the Conclusion implies the intention at the outset was reduce the number of questions and factors of the Brief COPE to improve utility in Mexican populations. Please reread carefully and be sure the paper relays and supports intentions.
Author Response
Reviewer 1:
Brief COPE short version (Mini-COPE): a proposal of item and factorial reduction in Mexican population aimed to reduce the Brief COPE, and by doing so simplify analysis of the instrument in Spanish-speaking populations.
There are many errors in word choice, grammar, flow, etc. The manuscript needs a careful read before moving forward. I will address some of the issues, but the list below is not intended to be exhaustive.
Abstract:
- Lines 18-20 – It’s not clear what the objective of the study is from this sentence. Is it to reduce the factors or replicate the reduction of factors?
Response: We clarified the objective of the study in the abstract section
- Line 33 – do you mean Spanish-speaking populations or a Spanish-speaking population?
Response: We corrected the term to “Spanish-speaking populations”
Introduction:
- The first sentence needs work. Line 39, which in turn, is considered…What is considered?
Response: We corrected the term to: “stress”
- The second sentence (line 41), doesn’t make sense.
Response: We corrected the phrase in order t make it clearer
- Line 44: Although these twoinstruments are similar and share assessment (or measurement) strategies, the Brief COPE is the most used in clinical and health contexts and measures a more significant number of strategies…
Response: We corrected the phrase as suggested
- Line 46: not in
Response: We corrected the word.
- Line 47: depending on the study. For example, Cooper … instrument to three dimensions.
Response: We corrected the phrase
The suggestions above should give the authors an idea of the types of word choice/grammar issues that need to be addressed.
Response: We proofread the whole paper to improve English grammar.
- One major issue with the summary of existing attempts to reduce factors in the COPE is the omission of the version of the COPE (situational vs dispositional) used in the studies and the impact that might have on results. See: Solberg MA, Gridley MK, Peters RM. The Factor Structure of the Brief Cope: A Systematic Review. Western Journal of Nursing Research. 2022;44(6):612-627. doi:10.1177/01939459211012044
Response: We added this important point at the end of the introduction section.
- Doesn’t the inability of the studies cited to “agree” on factors indicate that there may be differences in the populations being studied and the types of strategies they might use?
Response: We consider that this point can also modify the factorial reductions performed in other studies, therefore, we added it in the introduction section.
Subjects and methods:
- How were participants selected? How was socioeconomic level measured?
Response: In the section of subjects and methods we specified the target population and the sampling method.
Statistical Analysis:
- Read and edit for clarity and site SPSS.
Response: We edited this section.
Results:
- What does a “medium socioeconomic level” mean?
Response: We specified this point in the results section
- Be consistent with using EFA and CFA.
Response: We verified the correct use of each one of these acronyms.
- Line 128: 9, 10 are the incorrect references here
Response: We corrected the references
- Revise sentence lines 157-158 for clarity.
Response: We revised these sentences.
Discussion:
- I suggest reorganizing the Discussion. Specifically, perhaps switch paragraphs 1 and 2. Also, in doing so, revise the first sentence in the paragraph beginning Brief COPE… for clarity. It is too long and difficult to follow. There are other sentences that a very long and need work.
Response: We switched the paragraphs and revised the mentioned sentence and all long sentences in the discussion.
- The first sentence of paragraph beginning line 202 doesn’t make sense.
Response: We corrected the sentence
- Revise the concluding paragraph. Be clear about the version of the COPE used and the conclusions that can be drawn from the results.
Response: We revised the concluding paragraph and specified the version of the Brief COPE used.
Other issues:
In the Introduction there is talk of replicating previous findings, yet the Conclusion implies the intention at the outset was reduce the number of questions and factors of the Brief COPE to improve utility in Mexican populations. Please reread carefully and be sure the paper relays and supports intentions.
Response: We corrected this important point, clarifying in the introduction section the objectives of factorial and item reduction of the instrument, and the convergent and divergent validity of the factors obtained.
In this sense, the replication of previous factorial structures, although performed, we decided to do not included as objective, this considering that the replication was not performed for all the factorial reductions reported, and because we do not want to deviate the focus from the main objectives of the study.
Reviewer 2 Report
The English language use needs adjusting.
I have provided my native speaker's adjustments.
I ask that you check my suggested corrections and if you don't agree
please have another native speaker give advice.
Otherwise this is a good paper and should be published.

Author Response
Reviewer 2:
The English language use needs adjusting.
I have provided my native speaker's adjustments.
I ask that you check my suggested corrections and if you don't agree
please have another native speaker give advice.
Otherwise this is a good paper and should be published.
Response: We performed most corrections suggested by the reviewer
Reviewer 3 Report
Thank you for the opportunity to review the interesting article Brief COPE short version (Mini-COPE): a proposal of item and factorial reduction in Mexican population, which aims to reduce the existing Brief Cope and to evaluate the psychometric characteristics of the newly proposed shortened Mini-COPE to evaluate adaptive and maladaptive coping strategies in Spanish speaking population.
The authors of the study were able to meet these objectives in an appropriate manner.
In the introduction, they summarize relevant research and, based on this research, appropriately establish the need for the present study.
The statistical analysis itself is carefully conducted and clearly describes and presents the results.
The discussion redundantly repeats some information from the introduction. However, the interpretation of the findings is sufficient, although little reliance is placed on and discussion of the findings of relevant studies in this area. Strengths and weaknesses are adequately described.
However, there are several points that could be modified to meke the article more precise.
Intoduction
Although I realize that the scope of the introduction is largely limited by length, and the Brief COPE instrument is one of the well-known instruments for measuring coping strategies, the article would have benefited from providing at least a basic characterization of the instrument before the authors begin to write about its actual reduction. For the latter, it would then be good to mention why this instrument needs to be reduced.
Lines 43+44 - the most used instruments are the coping strategies inventory (CSI) , and the Brief COPE....
- In which setting are these studies among the most used? On what basis do the authors base their claim that these are the most used instruments?
Line 47 - The Brief COPE has been reduced in different dimensions depending of the study
- A correct usage of "depending" is with "on"
Line 56 - In addition, most of the reduced factors among the different populations studied are not similar among them.
- This sentence is hard to understand, among "what?".
- What are the most reduced factors?
Lines 62 -66 - give reference directly to relevant studies, not just the Review by Rodrigue et. Al (2022)
72-73 although the Brief COPE is already reduced,
- has been already reduced?
- which version the authors are talking about at this point, when they themselves state above that many attempts at reduction have been made.
2. Subjects and methods
What was the strategy used to select the participants of the study, was not targeted mailing only through social networks to reach only the circle of acquaintances of the research team? How was the balance of the sample ensured? There is a lack of information on who designed the questionnaire and whether (and how) its clarity was checked.
Although some of this information is provided in the study's limitations, it would be useful to mention it here.
In what language was the questionnaire itself and how was the presumed translation from the original instruments into Spanish done.
For the instruments used, it would be useful to describe them in more detail and to indicate what each range of scales measures and how the results were further treated for the purposes of this study. Are the scales provided validated for the Spanish language environment?
Results
What were the inclusion criteria for the research. Were any respondents excluded based on predetermined criteria? (e.g., completing the questionnaire in a very quick time, which could indicate mere "clicking off" responses and would not represent real data, etc.)
Discussion
Lines 168-173 seem redundant at this point, being merely a repetition of the information given in the introduction to the study. So are some of the statements in the previous paragraph. For the purposes of discussion, the old very brief summary serves as an introduction for the actual results.
Line 176 attempts to reduce this instrument [8]. , "this" is ambiguous in the context of the reference given in the brackets
190 - these emotional strategies (emotional and instrumental sup- 190 port) are slightly maladaptive - what do the authors use to explain these results?
Line 209 - do the authors know what the time saving is? It is significant enough that using only 2 factors is sufficient in terms of measurement accuracy.
At the very end of the study, in addition to describing the main findings, it would be useful to state how these findings can serve the professional community and in understanding coping strategies in general.
Overall, thanks to the authors for their efforts and research in the important area of coping strategies.
Author Response
Reviewer 3:
Thank you for the opportunity to review the interesting article Brief COPE short version (Mini-COPE): a proposal of item and factorial reduction in Mexican population, which aims to reduce the existing Brief Cope and to evaluate the psychometric characteristics of the newly proposed shortened Mini-COPE to evaluate adaptive and maladaptive coping strategies in Spanish speaking population.
The authors of the study were able to meet these objectives in an appropriate manner.
In the introduction, they summarize relevant research and, based on this research, appropriately establish the need for the present study.
The statistical analysis itself is carefully conducted and clearly describes and presents the results.
The discussion redundantly repeats some information from the introduction. However, the interpretation of the findings is sufficient, although little reliance is placed on and discussion of the findings of relevant studies in this area. Strengths and weaknesses are adequately described.
However, there are several points that could be modified to meke the article more precise.
Intoduction
Although I realize that the scope of the introduction is largely limited by length, and the Brief COPE instrument is one of the well-known instruments for measuring coping strategies, the article would have benefited from providing at least a basic characterization of the instrument before the authors begin to write about its actual reduction. For the latter, it would then be good to mention why this instrument needs to be reduced.
Response: We added a basic characterization of the instrument in the introduction section. The need to reduce the instrument is mentioned in lines 95-102 of the introduction section.
Lines 43+44 - the most used instruments are the coping strategies inventory (CSI) , and the Brief COPE....
- In which setting are these studies among the most used? On what basis do the authors base their claim that these are the most used instruments?
Response: We modified the redaction of the phrase.
Line 47 - The Brief COPE has been reduced in different dimensions depending of the study
- A correct usage of "depending" is with "on"
Response: We corrected the word.
Line 56 - In addition, most of the reduced factors among the different populations studied are not similar among them.
- This sentence is hard to understand, among "what?".
- What are the most reduced factors?
Response: We clarified the sentence
Lines 62 -66 - give reference directly to relevant studies, not just the Review by Rodrigue et. Al (2022)
Response: We added the specific references
72-73 although the Brief COPE is already reduced,
- has been already reduced?
- which version the authors are talking about at this point, when they themselves state above that many attempts at reduction have been made.
Response: We clarified this point, mentioning that the Brief COPE is a reduction of an original full COPE instrument of 60 items.
- Subjects and methods
What was the strategy used to select the participants of the study, was not targeted mailing only through social networks to reach only the circle of acquaintances of the research team? How was the balance of the sample ensured? There is a lack of information on who designed the questionnaire and whether (and how) its clarity was checked.
Response: We clarified the target population and the sampling method, and added information about the questionnaire designment.
Although some of this information is provided in the study's limitations, it would be useful to mention it here.
Response: We added this information in the methods section.
In what language was the questionnaire itself and how was the presumed translation from the original instruments into Spanish done.
Response: The original instrument is in English language; however, we used the Spanish translation performed by Morán, et al, 2010.
For the instruments used, it would be useful to describe them in more detail and to indicate what each range of scales measures and how the results were further treated for the purposes of this study. Are the scales provided validated for the Spanish language environment?
Response: We clarified this point in the methods section
Results
What were the inclusion criteria for the research. Were any respondents excluded based on predetermined criteria? (e.g., completing the questionnaire in a very quick time, which could indicate mere "clicking off" responses and would not represent real data, etc.)
Response: We clarified this point in the results section.
Discussion
Lines 168-173 seem redundant at this point, being merely a repetition of the information given in the introduction to the study. So are some of the statements in the previous paragraph. For the purposes of discussion, the old very brief summary serves as an introduction for the actual results.
Response: We moved and reduced this paragraph.
Line 176 attempts to reduce this instrument [8]. , "this" is ambiguous in the context of the reference given in the brackets
Response: We modified the phrase in order to make it clearer.
190 - these emotional strategies (emotional and instrumental sup- 190 port) are slightly maladaptive - what do the authors use to explain these results?
Response: We refer to the low but significant correlations obtained between this factor and anxiety, as mentioned 2 lines above: “Nevertheless, it is important to consider that the very low correlations observed with stress, depression and anxiety, reaching significance only with anxiety,”
Line 209 - do the authors know what the time saving is? It is significant enough that using only 2 factors is sufficient in terms of measurement accuracy.
Response: We consider that the accuracy remains when the researchers want to measure adaptive and maladaptive strategies, which are the most important strategies measured by the instrument in terms of clinical significance. Actually, the accuracy of these strategies is even higher, because 4 items per factor are included, instead the 2 items per subscale that are included in the Brief COPE.
Therefore, the reduction of the instrument implies saving time, accuracy in the measurements and clinical utility. We specified this in discussion.
At the very end of the study, in addition to describing the main findings, it would be useful to state how these findings can serve the professional community and in understanding coping strategies in general.
Response: We added some sentences specifying this point
Overall, thanks to the authors for their efforts and research in the important area of coping strategies.